# Pain(less) cleansing: Watching other people in pain reduces guilt and sadness but not shame

Konrad Bocian[1,2]*, Wieslaw Baryla[2]

**1** School of Psychology, University of Kent, Canterbury, United Kingdom, **2** Department of Psychology in Sopot, SWPS University of Social Sciences and Humanities, Warsaw, Poland

* K.Bocian-660@kent.ac.uk

## Abstract

Past research has shown that pain experience reduces feelings of guilt for earlier wrongdoings. In this paper, we aim to investigate whether watching other people in pain can reduce feelings of guilt. In Study 1 ($N = 60$), we found that participants' levels of guilt and sadness decreased after they watched a one-minute movie clip showing a painful medical procedure. Study 2 ($N = 156$), eliminated an alternative explanation in which pain observation but not the misattribution of unrelated excitation reduced guilt. Finally, in Study 3 ($N = 60$), pain observation lowered participants' feelings of guilt but not their feelings of shame. Overall, these results suggest that the guilt-reducing effect of pain may appear even without the actual experience of physical pain.

## Introduction

Pain is an undoubtedly aversive and unpleasant experience not only physically but also emotionally. Nevertheless, recent evidence from biological and psychological research suggests that pain might fulfil an important role in people's life. For example, studies have suggested that pain facilitates pleasure by providing contrast to pleasurable experiences [1, 2]. Similarly, researchers have found that pain serves as an indulging mechanism: people are more likely to choose to eat sweets and chocolate after a painful experience [3]. Moreover, relief from pain is rewarding and pleasurable because relief from pain and reward are processed similarly by the brain [4]. More interestingly, research has found evidence for a phenomenon called "moral masochism", which was coined by Sigmund Freud [5], who argued that suppression of guilt leads to a need for suffering. Even though empirical evidence that guilty people wish to suffer or want to be punished was inconclusive for many years [6] recent experimental studies have shown a direct association between pain (e.g., seeking self-punishment) and feelings of guilt.

In one study, participants who were led to believe that they financially harmed their partner later enforced more penalties on themselves [7]. In a different experiment, participants held their hands in ice water longer than did participants in the control group after remembering the last time they felt guilty. Additionally, the pain experience reduced their feelings of guilt [8]. A further experiment has shown that guilt, but not sadness, motivates people to punish themselves. Participants who recalled a guilt-inducing event self-administered significantly more intense electric shocks than did participants in the sadness-inducing event, whereas the latter did not differ from the control group [9].

**Data Availability Statement:** The data are openly available at https://osf.io/d4597/.

**Funding:** Preparation of this article was supported by the National Science Center (Poland) grant 2018/29/B/HS6/00658 (OPUS) (https://ncn.gov.pl/

?language=en), and by the Polish National Agency for Academic Exchange grant PPN/BEK/2018/1/00056 (Bekker Programme Scholarship) awarded to KB (https://nawa.gov.pl/en/). The funders had no role in study design, data collection and analysis, decision to publish, or preparation of the manuscript.

**Competing interests:** The authors have declared that no competing interests exist.

Based on the research above, we attempted to investigate whether feelings of guilt might be reduced without the actual experience of physical pain. Specifically, we aimed to examine if observing others' pain impacts guilt. Furthermore, we tested whether observing others' pain would affect sadness and shame as well.

## Social pain

Pain does not always have to be physical. We can experience pain without any direct somatic stimulation. We are all familiar with unpleasant situations (e.g., breakup, social exclusion, loss of a loved one) that feel like "pain" or a "wound", even if there was no physical harm. In social psychology, this type of "pain" is referred to as "social pain"—a signal that our social relations and attachment system, in general, are threatened [10, 11]. More importantly, past research has suggested that "social pain" triggered by social exclusion might be processed by some of the same neural circuitry that processes physical pain, see [12] for a review.

In a classic experiment conducted by Eisenberger, Lieberman, and Williams [13], participants' brains were scanned while the participants were playing a virtual ball tossing game in which they were ultimately excluded. Brain scans revealed a stronger activation in the anterior cingulate cortex (ACC) during the exclusion phase than in the inclusion phase of the game [13]. The pattern of neural activation for exclusion was very similar to that found in studies of physical pain, thus providing evidence that social and physical pain share a common neuroanatomical basis [14]. Even more robust evidence was found in a study in which participants were exposed to both painful physical stimulation and social threat [15].

The results demonstrated that when a feeling of rejection is powerfully elicited—by viewing a photograph of an ex-partner and thinking about being rejected—the affective component of pain (the experience of the unpleasantness) and the sensory-discriminative component of pain (the intensity of pain and its bodily location) became active [15]. Additional studies have shown that this neural pattern of activation holds when we experience social pain while being excluded in a ball tossing game and when we are witnessing the social pain of others being excluded [11].

Overall, the evidence we reviewed suggests that people may feel a pain event without directly experiencing psychical pain. In fact, we have ample evidence showing that the vicarious experience of pain can be generated when people observe another's pain. Therefore, peoples' physiological response might be similar to, but qualitatively different from, physical pain, or people might report feeling pain themselves when seeing others in pain [16–19].

## Vicarious experience of pain

From a neurological standpoint, pain is multidimensional and covers three broad components: the sensory-discriminative component (the intensity of pain and its bodily location), the affective component (the experience of the unpleasantness) and the cognitive component [20, 21]. These components, due to the observation of another's pain, might be activated and may produce an experience similar to the experience of physical pain. For example, participants' hand muscles potentials are reduced when they see another's hand being hurt, but such inhibition in the sensory-discriminative component of pain was not found when they saw a needle penetrating a tomato or cotton swab moving over the hand. This muscle inhibition suggests activation of a pain resonance system that extracts essential sensory aspects of the other person's painful experience [22]. A different study found activity in the primary somatosensory cortex of participants who observed the needle penetrating the hand [23]. This evidence suggests that observing which part of the body was hurt and how intensely it was hurt might trigger vicarious pain in the observer via activation of the sensory-discriminative component of pain.

Vicarious pain experience can be triggered by the affective component of pain, too. Research has demonstrated that the affective component of pain was activated when participants were viewing facial expressions of pain [24]. Similar activation of the affective component of pain was found when participants observed a signal indicating that their loved one receives pain stimulus [25]. Moreover, we have evidence suggesting that the activation of affective component of pain can be modulated by a wide range of personal and contextual factors, such as the moral behaviour of the person in pain [26] his or her medical expertise [27] or pain resulting from medical treatment [28].

Cumulatively, research on social and vicarious pain experience suggests that people might report feeling the pain without being physically hurt. Because past evidence has suggested that after the painful experience participants' levels of guilt decreased after the painful experience [8–9], we propose that observing another's pain might generate in the observer a superficially similar but qualitatively different kind of painful experience and therefore generate the guilt-reducing effect of pain.

## Research overview

We had two specific goals for devising this research. First, we aimed to investigate whether watching other people in pain would decrease only guilt or other negative emotions, such as sadness or shame. The second goal was to exclude an alternative explanation of the proposed effect in which guilt could be reduced by inducing intense arousal not related to pain (e.g., excitement). We tested our predictions in three studies.

In Study 1, participants first reminded themselves of an event from their life when they felt sad, guilty, or neutral. Afterwards, all participants watched a one-minute movie clip showing the painful blood collection procedure. We investigated whether levels of participants' guilt and sadness would decrease after they watched another person in pain. In Study 2, we used two additional movie clips—one arousing and one neutral—to test whether guilt might be lowered by misattributing unrelated excitation. In Study 3, we examined whether participants' shame would decrease after watching another people in pain. We first manipulated whether participants reminded themselves time in their life when they felt guilty or ashamed, and then they all watched a movie clip from Study 1. In the end, we tested whether participants' feelings of guilt and shame were affected by observing another person in pain.

In this article, we report all measures, all manipulations, and any data exclusions. Any additional measures not included in the main analyses are reported in the S1 File. In no study was additional data collected after seeing the results. The Ethics Committee Chair of Ethical Review Board at SWPS University of Social Sciences and Humanities has approved all studies (decision number: WKE/S 1/III/26). We obtained the written consent from participants via Sona platform. Participants proceeded to the study only when they agreed to participate in the study after reading the description of the study, time involved, and data policy. The data that support the findings of this study are openly available at https://osf.io/d4597/.

## Study 1

In Study 1, we investigated whether observing another's pain would reduce participants' guilt. We asked participants to describe an event from their past when they felt most guilty or sad or went to the grocery shop and then we asked them to watch a short movie clip about a blood collection procedure. We tested whether levels of guilt and sadness would decrease after watching the movie clip. We predicted that observing another's pain would reduce feelings of guilt. However, because by contrast to past studies [9] participants passively observed another's pain instead of effectively inflicting the pain on themselves, we did not have any specific predictions about sadness.

## Method

**Participants and procedure.** Sixty undergraduate students (47 women; mean age = 26.61 years, *SD* = 5.60) were recruited to participate in a study about self-reflection and negative life events in exchange for course credit. Although we did not use power analysis for sample size estimation, based on a sensitivity power analysis, this sample size provides 0.80 power for detecting an effect size of $f^2$ = .21.

Participants were individually taken to separate cubicles where they were handed instructions, a sheet of paper and an envelope. First, we informed participants that we were interested in how self-reflection can help people cope with negative life events. Then, based on a random assignment, the participants were asked to write about the last time they felt most guilty (*N* = 20) or sad (*N* = 20), or the last time they went to a grocery shop which served as a control condition (*N* = 20). To strengthen manipulation and participant involvement during the writing task, we asked the participants to answer questions about the emotional complexity of the event–what happened, how they felt and the consequences of their actions. When participants finished, they were told to put their stories in an envelope, seal it and notify the experimenter.

Next, participants rated their feelings of guilt and sadness. Afterwards, we asked the participants to watch a one-minute movie clip about a blood collection procedure (see the S1 File for more information). In the movie, participants observed the forearm of an unknown person and another person (presumably a doctor, however this information was unknown) trying to insert a needle into the vein of the examined person. After watching the movie, participants rated again how guilty and upset they felt, completed a demographic questionnaire and watched a funny movie clip, which was aimed to improve their mood. In the end, we debriefed participants and asked if they were aware of the real purpose of the experiment (none of the participants guessed the true aim of the study).

**Measures. Guilt and sadness** were measured with two items from the 20-item Positive and Negative Affect Schedule [29]. By using a scale from 1 = *very slightly or not at all* to 5 = *extremely*, participants indicated to what extend they felt guilty and upset.

## Results

**Manipulation control.** Using a one-way ANOVA, we tested how guilty and sad participants felt after the writing task in the guilt and sadness conditions. We found a significant effect of the manipulation for guilt $F(2, 57)$ = 8.38, $p$ = .001, $\eta^2$ = 0.23. Pairwise comparison showed that after the writing task, participants felt more guilt in the guilt condition (*M* = 2.50, *SD* = 1.19) compared to the control group (*M* = 1.25, *SD* = 0.55, $p$ = .001), but felt as much guilt as the participants in the sadness condition (*M* = 2.20, *SD* = 1.15, $p$ = .808). We also found a significant effect of manipulation for sadness $F(2, 57)$ = 4.08, $p$ = .022, $\eta^2$ = 0.12. In the sadness condition, participants felt more sadness (*M* = 2.40, *SD* = 1.10) than the participants in the control group (*M* = 1.50, *SD* = 0.89, $p$ = .023), but there was no difference in feelings of sadness whether the participants were in the guilt or sadness condition (*M* = 2.15, *SD* = 1.09 vs. *M* = 2.40, *SD* = 1.10, $p$ = 1.00).

**Guilt and sadness.** To test the hypothesis that guilt may be reduced by observing another's pain, we conducted a mixed ANOVA analysis in a 3 (Condition: Guilt vs. Sadness vs. Control) x 2 (Guilt: Time 1 vs. Time 2) design with the first factor as between-subjects and the second factor as within-subjects. This analysis revealed a significant main effect of the condition in which the levels of guilt were higher in the guilt and sadness conditions than they were in the control condition $F(1, 57)$ = 5.78, $p$ = .005, $\eta_p^2$ = .17. The main effect of time was also significant in that the guilt level decreased after the pain activation $F(1, 57)$ = 29.29, $p < .001$, $\eta_p^2$ = .34. More importantly, an interaction between condition and time was also significant $F$

(2, 57) = 7.02, $p$ = .002, $\eta_p^2$ = .20. Pairwise comparisons showed that in the guilt condition, participants experienced a significant reduction in guilt: Time 1 ($M$ = 2.50, $SD$ = 1.20) versus Time 2 ($M$ = 1.50, $SD$ = 0.69), $t(19)$ = 4.60, $p$ < .001, $d$ = 1.02, 95% CI [0.34, 1.66]. In the sadness condition, the effect of guilt reduction was also significant: Time 1 ($M$ = 2.20, $SD$ = 1.15) versus Time 2 ($M$ = 1.55, $SD$ = 0.89), $t(19)$ = 3.16, $p$ = .005, $d$ = 0.70, 95% CI [0.20, 1.18]. For the control condition, the effect was not significant ($p$ = .577).

To test whether sadness was also affected by pain manipulation, we conducted a similar analysis but with sadness as the repeated measure. This analysis showed no main effect of the condition ($p$ = .149) nor a main effect of time ($p$ = .241). However, the interaction between the condition and time was significant $F(2, 57)$ = 4.33, $p$ = .018, $\eta_p^2$ = .13. Pairwise comparison showed only one significant effect for the sadness condition in which the participants' level of sadness decreased after the pain manipulation: Time 1 ($M$ = 2.40, $SD$ = 1.10) versus Time 2 ($M$ = 1.90, $SD$ = 1.12), $t(19)$ = 2.52, $p$ = .021, $d$ = 0.45, 95% CI [0.19, 1.07]. The guilt and control condition comparisons were nonsignificant ($p$ = .789) and ($p$ = .104 respectively). Fig 1 presents the main results for the induced and measured emotions for guilt, sadness and control conditions.

## Discussion

The results of Study 1 provided initial support for our hypothesis that guilt would be reduced by observing another's pain. In accord with our predictions, participants' guilt decreased after the participants observed another person in pain. In the control condition, we did not find any change in the levels of guilt or sadness. However, we did observe guilt reduction in the sadness condition. On the one hand, this finding is likely because, in addition to inducing sadness in this condition, we also induced guilt. On the other hand, similar levels of sadness and guilt observed in both conditions might suggest some issues with the validity of manipulation used in Study 1. Nevertheless, the results of Study 1 suggest that the guilt-reducing effect of pain found in the past research [8–9] might be triggered by merely observing another person in pain.

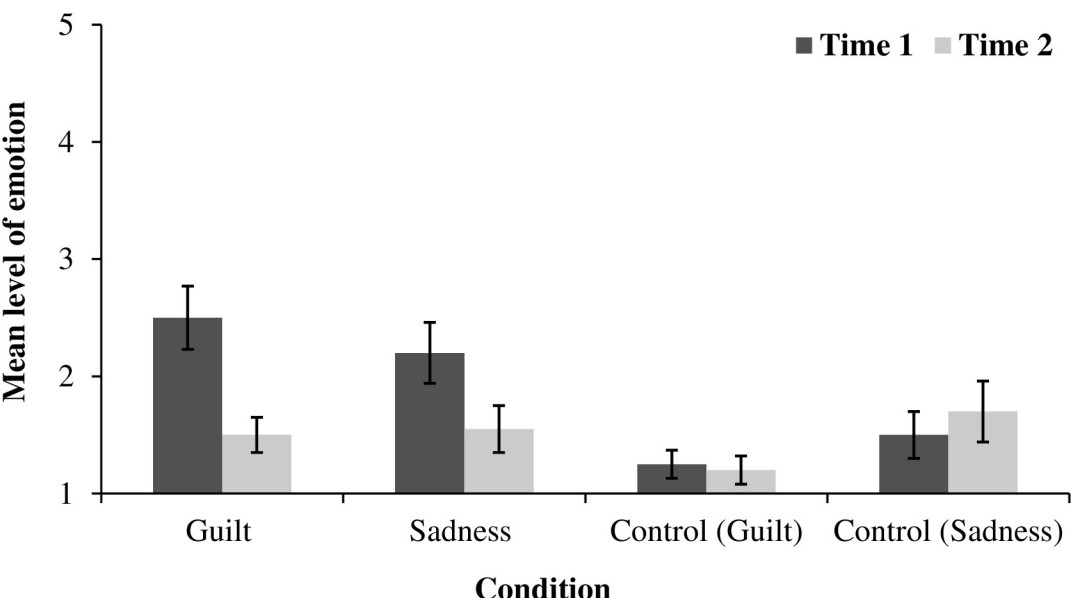

**Fig 1. The impact of watching the pain on recalled and measured emotions for experimental conditions and for the control condition in which emotions were measured but not induced.** Error bars represent standard error.

In contrast to past evidence that sadness does not motivate people to seek self-punishment [9] we found that observing another's pain affected participants' feelings of sadness. This result is somewhat unexpected and challenging to explain. On the one hand, unlike past research [9] participants in our study could not decide how much pain they wanted to inflict on themselves. Instead, they had to observe another person in pain. Thus, this difference might explain why the participant's sadness was affected by pain observation manipulation. On the other hand, sadness might fade away with time or because of the intense arousal triggered when participants were watching another person in pain. We sought to examine this alternative explanation in Study 2.

## Study 2

In Study 2, we sought to replicate the results of Study 1 by using a larger sample and different emotion measurement. Moreover, we tested an alternative explanation of the impact of observing another's pain on guilt reduction. Specifically, we investigated an idea that unrelated to the pain observation arousal evoked by intensely exciting stimuli instead of a painful one, might later reduce guilt. Much research indicates that the misattributing arousal influences emotional experience [30], sexual attraction [31] and evaluation of social events [32]. Thus, one can assume that guilt would be overwritten by intense arousal evoked by observing another person being hurt, which would reduce guilty feelings. However, if the association between guilt and observing another's pain is specific, we should find that guilt decreases only after the experience of vicarious pain but not after arousal activation. Another explanation of the effect found in Study 1 might be that both guilt and sadness fade over time.

Based on a random assignment, we asked participants to recall a time when they felt guilty or sad, and then we asked them to either watch the painful movie clip from Study 1, the exciting movie clip or the neutral movie clip ($N \sim 26$ for each group). We examined whether participants' feelings of guilt and sadness changed after they watched a specific movie clip. We assumed that guilt would be reduced after the observing another's pain but not after observing an exciting experience or the time passage produced by the neutral video.

### Method

**Participants and procedure.** One hundred fifty-six undergraduate students (90 women; mean age = 23.01 years, $SD$ = 4.05) were recruited to participate in a study about self-reflection and negative life events. Based on a sensitivity power analysis, this sample size provides 0.80 power for detecting an effect size of $f^2$ = .15.

Study 2 was a replication of Study 1 with two exceptions: First, we presented participants two additional conditions in which they watched a movie clip of either a man jumping from a crane on a bungee rope from the first person's perspective (the excitement condition) or a gnu grazing in a meadow (the control condition; see the S1 File for the results of the pilot study). Second, we changed the way guilt and sadness were measured to increase the external validity of this study.

**Measures. Guilt** was measured with two items: "guilt" and "remorse". Participants indicated to what extent they feel each emotion by using a 10-point scale from 1 = *I do not feel it at all* to 10 = *I feel it extremely strongly* (Time 1, α = 0.50, $M$ = 3.72, $SD$ = 2.13 vs. Time 2, α = 0.80, $M$ = 3.26, $SD$ = 2.27).

**Sadness** was measured with four items: "depression", "sadness", "misery", and "breakdown". Participants rated how they feel each emotion by using a 10-point scale from 1 = *I do not feel it at all* to 10 = *I feel it extremely strongly* (Time 1, α = 0.71, $M$ = 2.57, $SD$ = 1.53 vs. Time 2, α = 0.46, $M$ = 2.04, $SD$ = 1.08).

## Results

**Manipulation checks.** We performed a mixed ANOVA analysis in a 2 (Condition: Guilt vs. Sadness) × 3 (Movie Activation: Pain vs. Arousal vs. Control) × 2 (Emotions at Time 1: Guilt vs. Sadness) design with the first two factors as between-subjects and the second as within-subjects. The analysis showed the main effect of the emotions at Time 1, $F(1, 150) = 32.08$, $p < .001$, $\eta_p^2 = .18$. Generally, participants felt guilt more strongly ($M = 3.72$, $SD = 2.12$) than sadness ($M = 2.57$, $SD = 1.53$). However, this effect was limited by a significant interaction with the condition, $F(1, 150) = 42.96$, $p < .001$, $\eta_p^2 = .22$. Guilt was stronger in the guilt condition ($M = 4.54$, $SD = 2.06$) and weaker in the sadness condition ($M = 2.86$, $SD = 1.84$), $F(1, 154) = 28.58$, $p < .001$. In contrast, sadness was stronger in the sadness condition ($M = 3.04$, $SD = 1.63$) than in the guilt condition ($M = 2.12$, $SD = 1.29$), $F(1, 154) = 15.17$, $p < .001$.

**Guilt and sadness.** We analysed the data in a mixed ANOVA in a 2 (Condition: Guilt vs. Sadness) × 3 (Movie Activation: Pain vs. Arousal vs. Control) × 2 (Measured emotion: Guilt vs. Sadness) x 2 (Time of the measure: Time 1 vs. Time 2) design in which the first two factors were between-subjects and the two remaining were within-subjects. To facilitate the interpretations of the results we present the result of the interaction between condition, measured emotion and time of measure, $F(1, 150) = 5.49$, $p = .020$, $\eta_p^2 = .04$ (see the S1 File for full analyses).

Corroborating the results obtained in Study 1, guilt decreased from Time 1 to Time 2 ($M = 4.57$, $SD = 1.79$ vs. $M = 2.89$, $SD = 2.39$) in the guilt condition but only when participants watched another's pain, $t(27) = 3.01$, $p = .006$, $d = 0.67$, 95% CI [0.13, 1.21]. Watching an arousing scene (the excitement condition) or mundane one (the control condition) had no impact on guilt ($p = .582$ and $p = .574$ respectively). Sadness decreased from Time 1 to Time 2 in the pain condition ($M = 2.88$, $SD = 1.53$ vs. $M = 1.72$, $SD = 0.59$, $t(25) = 3.54$, $p = .002$, $d = 0.53$, 95% CI [0.02, 1.08] and in the control condition ($M = 3.30$, $SD = 1.56$ vs. $M = 2.34$, $SD = 1.34$, $t(25) = 2.36$, $p = .026$, $d = 0.43$, 95% CI [0.12, 0.98], but not in the excitement condition ($p =. 367$). The results are presented in Fig 2.

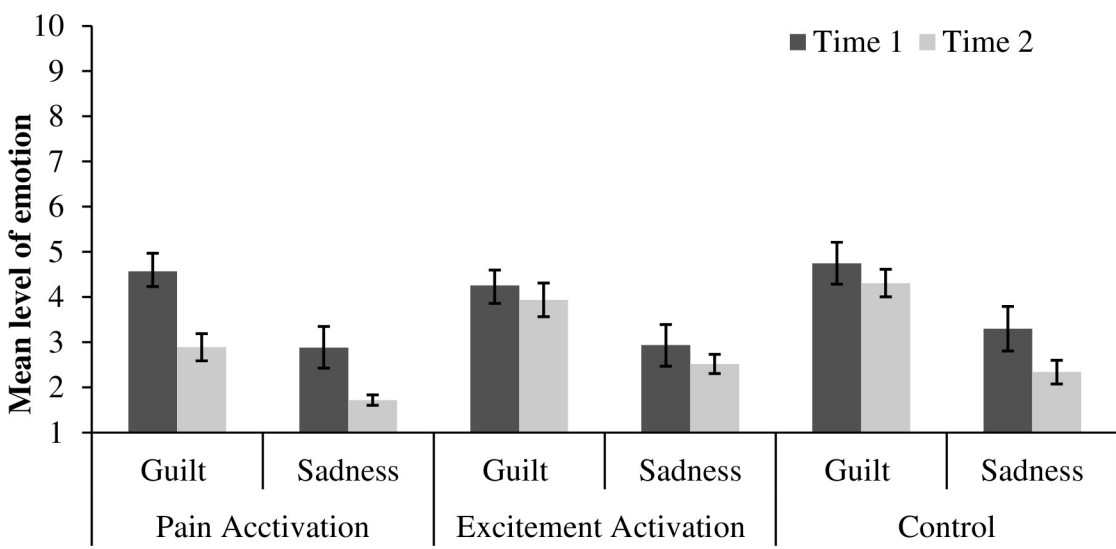

**Fig 2. The impact of watching the pain, excitement or monotony on recalled and measured emotions.** Error bars represent standard error.

## Discussion

The results of Study 2, with the help of a larger sample and different measure of emotions, replicated the results of Study 1 and eliminated alternative explanations of the guilt-reducing effects of the vicarious pain experience. Similar to Study 1, we found that guilt decreased after participants watched another person in pain. More importantly, we showed that neither the excitement activation nor the time passage influenced participants' feelings of guilt. Therefore, we ruled out the alternative explanation that guilt can be replaced and lowered by another consecutive emotion, such as excitement.

Corroborating the result of Study 1, we found that sadness was also reduced after the observation of another's pain. However, because we also observed a decrease of sadness in the control condition, we assume that sadness somewhat declined through the time passage, since the neutral movie was boring and mundane. We discuss more specifically how sadness could be affected by the vicarious pain manipulation in the general discussion.

## Study 3

So far, two studies confirmed that guilt might be reduced after observing another person in pain. In Study 3, we aimed to replicate this primary effect with additional extension. Specifically, we investigated whether observing another's pain would reduce shame, another emotion close to guilt.

At first glance, guilt and shame seem to share the same properties. Both emotions are negative in their valence, and both share internal (vs. external) elements of blame and activate a need for coping [33]. Moreover, guilt and shame frequently co-occur in many situations [34]. They both are associated with social and moral norm violations [35] and arise from self-relevant failures and transgressions [36]. However, much of the research emphasizes differences in the phenomenology of shame and guilt [37–41].

Cross-cultural studies have found that participants reported that their shame experiences were elicited more often by other people or external sources than by guilt experiences [42], see also [43]. Further, research has demonstrated that when people experience shame, they tend to focus on themselves, but when they experience guilt, they tend to focus on their behaviour [44]. Finally, shame arises when adverse events are attributed to one's stable, global self while guilt arises when adverse events are attributed to unstable, specific aspects of the self [45]. Moreover, guilt evolved from a need to avoid harming others [33] while in contrast, shame is a self-focused emotion that is related to needs to prove oneself as acceptable or desirable to others [46].

Research concerning the neural activity of brain regions shows another difference between shame and guilt. Experience of shame is associated with increased activation in the ACC [47, 48] the same brain region, which is active when people experience physical pain, social pain and when they observe pain in others. Furthermore, research has suggested a specific link between social pain and shame, because social exclusion increases shame, and this link is mediated by feelings of being devalued [49]. These results correspond with cross-cultural research which has shown that shame is strongly associated with tracking the magnitude of the devaluative threat in others [50] or evidence that people who have experienced shame prefer to be together with others over being alone [51].

By contrast, guilt is processed by a region of the right orbitofrontal cortex (OFC) and the paracingulate dorsomedial prefrontal cortex (DMPFC). Specifically, research has found evidence that the more participants reported being prone to experiencing guilt, the more they recruited the right OFC. More importantly, no other region in the brain was correlated with the dispositional measure of guilt, while the OFC and the DMPFC were inactive when other

emotions, such as shame or sadness, were induced [52]. Moreover, people feel guilt when they exclude another person [53].

Overall, presented evidence suggests that experiences of shame, physical pain, and social pain share the same neural basis because they are all associated with increased activation in the ACC [11–15, 47, 48]. Therefore, it is plausible that shame levels would remain intact after pain observation because the same brain regions process both shame and observation of another person in pain (e.g., ACC). On the other hand, discovered in Study 1 and Study 2 the guilt-reducing effect of pain observation may occur because different brain regions process guilt (e.g., OFC) and these brain regions are not involved in pain processing [52].

## Method

**Participants and procedure.** We recruited sixty undergraduate students (42 women; mean age = 21.60 years, $SD$ = 1.75) to participate in a study about self-reflection and negative life events. Based on a sensitivity power analysis, this sample size provides 0.80 power for detecting an effect size of $f^2$ = .12.

**Procedure and materials.** Study 3 was a replication of Study 1 with two exceptions: First, we induced only guilt and shame. Second, we used different manipulations of emotion induction. Specifically, we used a procedure tested in a previous experiment by Wagner [52]. Thus, based on a random assignment, we asked participants to recall and describe as precisely as possible a particular event from their life when they either: 1) acted against their own rules, and they caused harm to another person by their fault (the guilt condition, $N$ = 30) or 2) their reputation was damaged, and they could not to change the negative impression that they caused (the shame condition, $N$ = 30).

We used this specific manipulation because it has several advantages over the manipulation from Studies 1 and 2. First, the indirect procedure produces a way in which participants can interpret target emotions because situation descriptions ensure that participants share events that have the same basis. Second, it is difficult to explicitly distinguish between situations of shame and situations of guilt. Third, participants' ratings of target emotions are not biased because they are not mentioned during the manipulation [52].

**Measures.** Guilt and Shame was measured with two items. Participants indicated to what extent they feel guilt and shame by using a 100-point sliding scale from 1 = *I do not feel it at all* to 100 = *I feel it extremely strongly*.

## Results

**Manipulation control.** The analysis of variance in a mixed model with a 2 (Condition: Guilt vs. Shame) x 2 (Emotion at Time 1: Guilt vs. Shame) design with the first two factors as between-subjects and the second as within-subjects showed no main effects and a significant interaction between factors $F(1, 58)$ = 9.19, $p$ = .004, $\eta_p^2$ = .14. As expected, participants felt more guilty ($M$ = 58.10, $SD$ = 32.13) than ashamed ($M$ = 40.57, $SD$ = 36.33) in the guilt condition and more ashamed ($M$ = 55.30, $SD$ = 28.87) than guilty ($M$ = 46.43, $SD$ = 29.09) in the shame condition.

**Guilt and shame.** To facilitate interpreting the results, we analysed the effects only for activated and measured emotions (see the S1 File for full analyses). For the guilt condition, we analysed only levels of guilt at Time 1 and Time 2, whereas for the shame condition, we analysed levels of shame at Time 1 and Time 2. The results are presented in Fig 3.

Corroborating the results from Studies 1 and 2, guilt decreased from Time 1 to Time 2 ($M$ = 58.10, $SD$ = 32.13 vs. $M$ = 17.80, $SD$ = 22.49) when participants watched another's pain, $t$(29) = 7.17, $p$ < .001, $d$ = 1.15, 95% CI [0.61, 1.70]. However, congruently with our predictions when participants watched another's pain, their levels of shame remained the same.

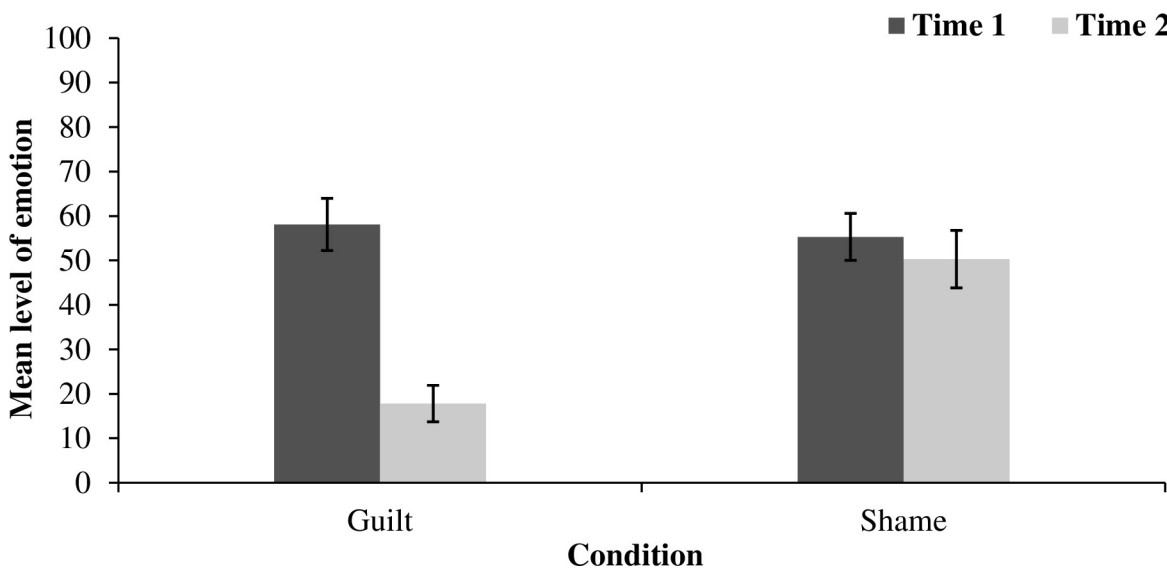

**Fig 3. The impact of watching the pain on recalled and measured emotions.** Error bars represent standard error.

Time 1, *M* = 55.30, *SD* = 28.86 versus Time 2, *M* = 50.30, *SD* = 35.45, *t*(29) = 0.93, *p* = .326.

## Discussion

The results of Study 3 corroborated findings from Studies 1 and 2 and provide additional evidence that watching other people in pain reduces feelings of guilt. Additionally, we found that shame remained intact after participants watched another person in pain. This pattern of results suggests that watching other people in pain influences levels of guilt and shame differently when a target emotion is recalled. The results of Study 3 also suggest that shame, as opposed to guilt, might show weaker associations with the vicarious pain experience.

## General discussion

In this paper, we sought to contribute to the scarce research on the guilt-reducing effect of pain [2, 3, 7–9] by finding evidence for a specific association between guilt and mere observation of others in pain. For the first time, we systematically showed that guilt could be reduced after an unpleasant experience triggered by observing others being hurt. In three studies, we found that induced guilt decreased after participants watched another person in pain. Additionally, in Study 2, we ruled out the alternative explanation showing that guilt did not decrease after the arousing experience. In Study 3, we found evidence that shame does not decrease as strongly as guilt after watching other people in pain. We assumed that observing others being hurt might trigger the guilt-reducing effect of pain because feeling and observing pain activates the same components of the pain system [22–26]. In three experiments, we found evidence corroborating that watching another person in pain reduces guilt.

## Theoretical contribution

The original contribution of these studies to the current literature is threefold. First, we contribute to the past research on the guilt-reducing effect of pain by presenting a novel approach to studying how pain observation affects guilt [8–9]. Second, in Studies 1 and 2, we found evidence that sadness also decreased after observing another's pain. Research by Inbar [9] showed that guilt motivates people to self-punishment more strongly than sadness. In the authors'

study, in all three conditions (guilt, sadness, and control), participants inflicted electroshocks on themselves, but in the guilt condition, participants gave themselves stronger shocks than in the other two conditions. The analysis of guilt at Time 1 and Time 2 showed that stronger shocks were associated with more alleviation of feelings of guilt [9]. Unfortunately, there is no information about levels of sadness and the impact that electric shocks had on this emotion.

It is at least plausible that sadness is also affected by painful experience but less strong than guilt. We found some support for this hypothesis in Study 1. Both guilt and sadness decreased after pain observation, but the size of the effect was more than twice the size for guilt ($d = 1.02$) than for sadness ($d = .45$). However, in Study 2, sadness decreased in both the pain and the control groups, thereby possibly implying that sadness wears off over time. More research about sadness in the context of self-punishment and the experience of pain observation is needed.

Finally, we found evidence implying that shame was unaffected by the experience of pain observation. We are aware that this result is thought-provoking and seems unexpected because guilt and shame are strongly bonded. However, past research has suggested that guilt and shame differ from each other because of the sources that trigger both emotions: external versus internal [42, 43], what people focus on when they feel these emotions: on the self versus on the behaviour [44]; and whether the emotion arises from events attributed to global or specific aspects of the self [45]. More importantly, both pain experience and shame are associated with activation of the ACC [47, 48] while guilt is processed by the OFC and the DMPFC brain regions [52].

Relying on these premises, on the one hand, we assumed that watching other people in pain would recruit the ACC, which past studies have proposed to be an important cortical brain region responsible for pain perception [54–56]. On the other hand, shame also recruits the ACC [48], therefore, we assumed that shame induction would recruit the ACC and further pain observation sustained this activation. Obviously, these results should be treated with great caution because we did not test activation of the ACC and other brain regions directly involved in guilt and shame processing.

## Limitations and further directions

We recognize that these studies have some limitations that might warrant future research. For example, in all experiments, we used the same manipulation of the pain observation. Another limitation concerns the fact that even though neuropsychological studies support our assumptions about the underlying psychological mechanism connecting guilt and pain observation, we did not test these assumptions directly. Therefore, future research could examine the intensity of activation of brain areas involving the sensory-discriminative and the affective component of pain shortly after guilt and shame induction and then during and after watching another person in pain. This analysis could contribute to a better understanding of how pain and observation of pain help people manage feelings of guilt and shame. Moreover, because of conceptual confusion that often occurs between shame and guilt, it is possible that the method used in Study 3 to measure these emotions did not adequately distinguish between these two constructs. Finally, samples used in our studies and the effects were small, thus warranting future research on larger samples.

## Conclusion

By systematically examining whether watching other people in pain reduces feelings of guilt, this research provides additional support for the theories that argue that guilt leads to a need for suffering [5–9]. The results of this research suggest that the guilt-reducing effect of pain occurs even without the experience of physical pain. Therefore, we conclude that merely observing other people in pain is sufficient to soothe our feelings of guilt.

## Supporting information

**S1 File.**
(DOCX)

## Acknowledgments

The authors would like to thank Michal Winnicki and Patryk Zamulski for their help in data gathering and Aleksandra Cichocka and Brianna Beck for their comments on an earlier draft of this manuscript.

## Author Contributions

**Conceptualization:** Konrad Bocian, Wieslaw Baryla.

**Data curation:** Konrad Bocian, Wieslaw Baryla.

**Formal analysis:** Konrad Bocian, Wieslaw Baryla.

**Funding acquisition:** Konrad Bocian.

**Investigation:** Konrad Bocian, Wieslaw Baryla.

**Methodology:** Konrad Bocian, Wieslaw Baryla.

**Project administration:** Konrad Bocian.

**Resources:** Konrad Bocian, Wieslaw Baryla.

**Supervision:** Wieslaw Baryla.

**Validation:** Konrad Bocian.

**Visualization:** Konrad Bocian.

**Writing – original draft:** Konrad Bocian, Wieslaw Baryla.

**Writing – review & editing:** Konrad Bocian, Wieslaw Baryla.

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
