## [Decision Letter · Decision Letter 0]

17 Jul 2020

PONE-D-20-01585

Pain(less) Cleansing: Vicarious Pain Experience Reduces Guilt and Sadness but not Shame

PLOS ONE

Dear Dr. Bocian,

Let me start by apologizing for the delay in sending you this action letter. I was contacted by the editorial office on August 7^th^ 2020 with the request to take over the handling of your manuscript as academic editor. After agreeing, I saw that the previous academic editor already secured one review and made a huge effort to find a second reviewer. However, the second review did not materialize. Fortunately, I could consult a colleague whose expertise is related to the topic of your paper. This allowed me to obtain a second review at very short notice.

Both reviewers agree that there is merit in your work but recommend a major revision before the paper can be accepted. Hence, I decided to not let you wait any longer and to send you the reviews with the request to submit a revised version of your paper that takes the comments of the reviewers into account. The reviewers provide many helpful suggestions on how you could improve your manuscript by changing the build up of your arguments or by resolving ambiguities and conceptual issues. Do not feel obliged to follow every single suggestion of the reviewers but please do use their comments in a constructive way with the aim of creating a drastically improved version of your paper.

I did notice, however, that the raw data of your studies are not available in the supplementary materials. It is standard policy of PLOS ONE that data are made publicly available unless there are valid reasons for not doing so (see https://journals.plos.org/plosone/s/data-availability ). Please consult our guidelines and make the data available in line with those guideline or inform us why this is not possible.

A final small comment related to the power issue raised by Reviewer 2: please delete the phrase that you did not calculate power “because it was not required back then”. It suffices to say that you did not calculate power before the start of the study.

Once more, on behalf of our journal, I apologize for the delay in taking action.

We look forward to receiving your revised manuscript.

Kind regards,

Jan De Houwer

Academic Editor

PLOS ONE

Journal Requirements:

2. Please update your Methods section to indicate whether your ethics committee approved the consent procedures (or lack thereof) in the study.

Your ethics statement must appear in the Methods section of your manuscript. If your ethics statement is written in any section besides the Methods, please move it to the Methods section and delete it from any other section. Please also ensure that your ethics statement is included in your manuscript, as the ethics section of your online submission will not be published alongside your manuscript.

Reviewers' comments:

Reviewer's Responses to Questions

**Comments to the Author**

1. Is the manuscript technically sound, and do the data support the conclusions?

Reviewer #1: Partly

Reviewer #2: Yes

2. Has the statistical analysis been performed appropriately and rigorously? 

Reviewer #1: I Don't Know

Reviewer #2: Yes

3. Have the authors made all data underlying the findings in their manuscript fully available?

Reviewer #1: Yes

Reviewer #2: Yes

4. Is the manuscript presented in an intelligible fashion and written in standard English?

Reviewer #1: No

Reviewer #2: No

5. Review Comments to the Author

Reviewer #1: Please see attached document for comments

Reviewer #2: Thank you for receiving the opportunity to review the manuscript entitled “Pain(less) Cleansing: Vicarious Pain Experience Reduces Guilt and Sadness but not Shame” which presents a set of interesting experimental studies aimed at testing whether observing someone else’s’ pain reduces emotions of guilt in the observer. I was intrigued by the topic of the research and the rigorous approach of several experimental studies to better understand the underlying mechanism of the main finding. However, there are still several methodological and conceptual concerns that I would like to see addressed before I can recommend this manuscript for publication.

Main points:

1. One of the main methodological concerns is the sample size of each study. The authors provide post hoc power calculations, but it is unclear how they arrived at these numbers or what they mean. Please provide a more detailed account of the power that was achieved by the current sample per study. Please follow the steps outlined by Daniel Lakens here and explain how likely it was to observe a significant effect, given your sample, and given an expected or small effect size and report all parameters entered in G*Power: http://daniellakens.blogspot.com/2014/12/observed-power-and-what-to-do-if-your.html.

a. It is a bit unclear why the authors report f effect sizes when their main analyses are repeated measures ANOVAs.

2. A conceptual issue relates to the idea that “mere observation of others in pain produces a full experience of pain in the observer” (p. 4). That statement seems hyperbolic and scientifically incorrect based on the research cited. It is true that observing someone else in pain activates similar neural regions than direct experience of pain, but it is also clear that the actual experience of observing someone else in pain and experiencing pain oneself is qualitatively quite different. For instance, no one would confuse whether they experienced pain themselves or they saw someone else experiencing pain. Moreover, the authors did not assess in any way what emotional reaction the participants had to the video of someone else in pain, how “painful” they experienced watching it, or how they were affected by it. In my mind, these are crucial variables to test the hypothesis the authors put forward. Could the authors comment?

a. Similarly, the authors keep speaking of “experiencing vicarious pain” without assessing it and is therefore speculative. I would prefer if the authors change this to “observing someone else in pain” or “vicarious pain perception” throughout the whole manuscript and in the title as this more clearly describes the manipulation which was employed.

3. There are several grammatical and language errors in the text, and the manuscript would profit from professional academic proofreading.

Minor points:

1. P. 2 “pain observation affected participants’ feelings of guilt” Please state the direction of the effect

2. P. 6 “shame and pain experience share the same neural underpinnings” What does that mean and why does that mean it would not be affected?

3. P. 7 “would be observed after watching an exciting and neutral movie clip” Why not use an arousing negative clip as a comparison? It would make the claim stronger that the guilt reduction is in response to pain alone.

4. P. 8 “asked to write about the last time” How long were the participants required to write? Did all participants write for the required time?

5. P. 8 Please include a paragraph on the actual design of the study. All studies employed (at least in part) a between-subjects design yet this is not explicitly stated anywhere. Also, the n of each group is not stated or how participants were randomized. Lastly, please refer to between-subject levels as “groups” rather than “conditions” throughout the manuscript.

6. P. 9 “they were told to put their stories in an envelope” What was the cover story and what did the participants think would happen with the stories? Were they told that they were kept confidential?

7. P. 9 “were measured with two items” Please state the exact wording, scale and labels of the items used.

8. P. 9 Please include a separate paragraph one the exact statistical analyses that were carried out per study, including the dependent and independent variables, their levels and the used alpha-level.

9. P. 10 “we conducted an ANOVA” Please specify what type of ANOVA (in this case a repeated measures ANOVA)

10. P. 11 “the participants’ level of sadness decreased after the pain manipulation” This is a very surprising finding and definitely requires further elaboration in the discussion. Earlier research shows that watching someone else in pain elicits unpleasant emotions and it is likely that one would feel sad for someone else in pain (especially with high empathy levels). So how do you explain a reduction in sadness when observing someone else in pain?

11. P. 11 In general, there are quite a few post-hoc comparisons. Did the authors employ any post-hoc corrections such as Holm-Bonferroni to correct for multiple testing? This is especially relevant since the studies are likely underpowered.

12. P. 11 “apart from inducing sadness in this condition, we also elicited guilt” Please elaborate. What do you mean?

13. P. 11 “(Bastian et al., 2011 (…)” bracket missing

14. P. 13 “We changed the way guilt and sadness was measured” Why did the authors choose to change their measurements throughout the studies? If they want to present it as a replication of Study 1 it seems counterintuitive to change the methodology. Please provide arguments for doing so.

15. P. 13 “was measured with four items” How were these items combined in a single outcome variable? Since this is an entirely new measure, please provide psychometric properties (e.g. Cronbach’s Alpha).

16. P. 14 Did you assess perceived pain intensity or empathy levels at any point?

17. P. 16 “past studies proved” Please avoid the word “proved” in scientific writing.

18. P. 18 “Finally, using Bayesian interference” The authors introduce a whole new set of statistical analyses here. Please be consistent throughout your studies in the analyses employed and if not, provide a clear rationale for changing method of analysis.

19. P. 22 “might help people deal with aversive feelings of guilt.” The notion that people watch violent shows because of feelings of guilt is highly speculative and far-fetched. I would reduce this last paragraph. Instead, I would be interested in practical or clinical implications of this research as these are currently missing in the discussion and introduction.

6. PLOS authors have the option to publish the peer review history of their article (what does this mean?). If published, this will include your full peer review and any attached files.

Reviewer #1: No

Reviewer #2: No

---

## [Author Response · Author response to Decision Letter 0]

5 Sep 2020

Dear Prof. De Houwer, 

We thank you very much for your helpful decision letter concerning our paper and for securing such thoughtful reviews. We attach a rewritten and reframed paper that follows all reviewers’ suggestions. To facilitate your work, we have pasted the reviewers’ comments, and we respond to all suggestions and comments in turn.

Please let us know if you have any questions at all concerning this submission. Thank you in advance for your consideration. 

Yours sincerely,

Konrad Bocian and Wieslaw Baryla

RESPONSES TO ISSUES RAISED BY EDITOR:

1. Editor wrote: “I did notice, however, that the raw data of your studies are not available in the supplementary materials. It is standard policy of PLOS ONE that data are made publicly available unless there are valid reasons for not doing so (see https://journals.plos.org/plosone/s/data-availability ). Please consult our guidelines and make the data available in line with those guideline or inform us why this is not possible.”

We made the raw data available under this link: https://osf.io/d4597/?view_only=9943b05ae9c649ba8d44a65fd5f2dbf1. Also, on p. 7 we write: 

The data that support the findings of this study are openly available at https://osf.io/d4597/?view_only=9943b05ae9c649ba8d44a65fd5f2dbf1. 

2. Editor further wrote: A final small comment related to the power issue raised by Reviewer 2: please delete the phrase that you did not calculate power “because it was not required back then”. It suffices to say that you did not calculate power before the start of the study.

 We removed the phrase as suggested. 

RESPONSES TO ISSUES RAISED BY REVIEWER 1: 

Introduction 

The introductory paragraphs on page 3 lack a cohesive narrative that set the stage for the paper. For example, the first paragraph does not really a clear lead into the first sentence in paragraph 2, which talks about the posited links between guilt and physical and observed pain. I think that the information about the importance of the experience of pain as it is linked to pleasure could be condensed and clarified, and the link between pain and guilt should be elaborated a bit. Furthermore, the last sentence in the first paragraph (i.e., “…physical pain redeems our sins by reducing guilt after moral transgression…”) is unclear. Additionally, some readers may not be familiar with the concept of moral transgressions. Given the focus on guilt, I think it would be worthwhile to provide a clear definition of this construct and its purpose.

The definition of moral masochism is unclear. Furthermore, I was confused about the link between feelings of guilt and the desire for self-punishment. Although there is supporting evidence provided for this link, I think it would be helpful to provide a rationale hypothesized in the literature as to why people who feel guilt want to engage in self-punishment. This might be accomplished by providing a clearer definition of the theory of moral masochism. 

The introduction discusses studies linking pain and guilt vs. sadness, however there is no discussion of the studies exploring links between pain and guilt vs. shame. Given that shame is a construct explored in Study 3, I think it would be important to add studies exploring these links to the introduction. Furthermore, I think it would create a stronger narrative for the manuscript if the information about shame and guilt that is discussed in study 3 was moved to the introductory section.

In the last paragraph of the section titled “Moral Masochism”, the first sentence states “Based on the research above, we assume…”. This is unclear, as the previously discussed research talks about linking guilt and actual pain (not vicarious). I suspect this was an attempt to lead into the next paragraph, however this link is unclear. It may be helpful to use more hypothetical language here that emphasizes the fact that you are making a speculation and highlights the fact that you will explore evidence for such speculations in the next section. 

The first paragraph under the section “Vicarious Experience of Pain” discussing the association between experienced and vicarious is pain lacks clarity. I also think it would be helpful to provide an example of a study here to support this link.

The discussion of the three neurological components of pain also lacks clarity. I think providing a clear definition of each component would help, particularly discussion of the sensory-discriminative component. 

In the second paragraph, it is unclear what is meant by “the participants froze their hand…”. I also found the link with inhibition unclear. It may be that the word “inhibition” is not the appropriate word here, or a need for a clearer explanation of this link. Also, it is noted that such inhibition was correlated with pain intensity but not empathy traits. It is not clear what the significance of this non-significant association is. It would be helpful to explain why this is important and also to provide examples of what empathy traits were examined or a definition for though who are not familiar with this construct. Additionally, an explanation of the purpose of the primary somatosensory cortex, sensorimotor activation, and their link to pain is necessary. The discussion of contextual factors should also be clarified with respect to how the studies discussed highlight why experienced and vicarious pain are similar. 

Given that guilt is important component of this paper, I think there needs to be a more detailed explanation of how and why the information discussed in this section supports your proposed link with guilt. 

We are thankful for these comprehensive and thoughtful suggestions proposed by the reviewer. In the revised version of the manuscript we followed most of the reviewer’s suggestions by substantially rewriting the whole introduction. To make our arguments stronger, we removed or rewrote the paragraphs that were not clear enough. In addition, we removed the section about moral masochism and added a new section about social pain. Thus, we present more evidence that supports our assumptions that people might feel or report feeling pain without being physically hurt. We would also like to stress that there are no studies exploring the link between guilt and pain apart from the studies we cite in the manuscript. Moreover, we are not aware of any studies exploring the link between shame, sadness and pain. 

Overview of Present Studies

The explanation of shame and its hypothesized non-significant association with vicarious pain is unclear. As previously noted, additional information about shame should be added to the introductory section.

An explanation for why three different studies were conducted should be added to this section. For example, it is later noted that Study 2 aimed at examining potential confounding factors. This could be added to the overview section to create a clearer narrative for reader. I also think it would be beneficial to clearly link each study to a specific hypotheses or goals. It might be helpful to incorporate the introductory paragraphs that are presented under each study into this section instead of later in the manuscript. 

The information about the specific procedures for each study could be condensed, given that this information and repeated in the procedure section in each study. 

On page 7 it states that information about data exclusion was included, however this is not this case. This information should be added to each study. 

As suggested by the reviewer, we condensed this section and rewrote some sentences to make the narrative clearer. 

Studies

Additional data available regarding participant demographics would be a useful addition if available. 

As previously noted, the introductory paragraphs found under each study may be better placed in the study overview. Also, these paragraphs discuss the procedure, which is redundant given that it is then presented in the methods section that follows. 

Study 1 

o Information about the procedure could be condensed by removing unnecessary additional information (e.g., that they put their stories in an envelope, sealed it, then notified the experimenter).

o It is noted that participants were asked if they were aware of the real purpose of the experiment, however, the findings of this were not reported. 

o Additional information should be given about the measure of guilt and sadness (e.g., which items, how were the questions phrased and rated).

o In the manipulation control section, it is noted that participants’ perceptions of how painful the blood collection was and the degree to which they identified with the protagonist. However, it was not explained why these analyses were conducted. Also, the information about these questions should be included in the measures section. 

o As noted in the permission guidelines, footnotes are not permitted. Also, in the footnote it mentions that the interaction remained significant when shame was controlled for, however, it is not clear why this analysis was conducted.

o Discussion – the first paragraph is redundant and does not offer enough interpretation of the results. Further, this section seems to be focused more on the links with sadness than with guilt. 

We followed all the suggestions proposed by the reviewer. We moved the information and the analyses about the perception of blood collection and the degree to which participants identified with the protagonist to the Supplement. We also removed the footnotes and moved the analyses presented in the footnotes to the Supplement.

Study 2 

o The introduction discusses “unrelated residual excitation”. It is not clear what this is referring to as it was not discussed in the introductory section. 

We removed the phrase “unrelated residual excitation” from the current version of the manuscript. Instead we write about “unrelated excitation” or more precisely “intense arousal”. 

o Why was the way guilt and sadness were measured changed for the purposes of this study?

We changed the way of measuring guilt and sadness to increase external validity of Study 2.

Study 3

o The introduction paragraph notes that this study aimed to replicate studies 1 and 2, however, this does not appear to be accurate as this study was focused on shame. 

o In discussing the differences between shame and guilt, it is noted that shame is more of a social emotion because shame is more often elicited by others or by external sources. This is somewhat misleading, as both guilt and shame can be elicited through interactions with others. It is also noted that shame affects the whole self, which is unclear. I believe this note is meant to highlight the fact that shame is focused on perceiving oneself as inherently bad, versus guilt, which is focused on the one’s behaviour as being bad. This should be clarified. It is also unclear how points c) and d) are different from points b). Also, the note about shame being self-oriented versus guilt being other-oriented is also misleading, as shame can also be externalized into behaviour directed towards others. It may also be useful to discuss how shame is maladaptive, whereas guilt is adaptive. 

o There is reference to numerous brain regions. The roles each of these regions should be explained, as many readers may not be familiar with them. Also, only the abbreviation for anterior cingulate cortex appears, without referencing what this abbreviation stands for. 

o A study is referenced that looked at ACC activation in response to embarrassment. However, shame and embarrassment are different constructs, thus it may be a good idea to reference a different study that looked specifically at shame. 

We are grateful for the reviewer’s suggestions. In the revised version of the manuscript, we followed them by substantially rewriting the whole introduction of Study 3. 

o The second paragraph on page 16 states that “past studies proved…”. It may be a good idea to select an alternative word, as we can never really “prove” things when it comes to research.

We removed the word “proved” from the current version of the manuscript. 

o The last part of the second paragraph on page 16 contains contradictory statements about the hypothesized relations between shame and pain. 

We removed this statement. 

o The procedure section does not mention that participants were asked to watch a video. 

Now we write: 

“Study 3 was a replication of Study 1 with two exceptions.”

o It is noted that only guilt was analyzed in the guilt condition, and only shame was analyzed in the shame condition. However, the rationale for this decision is not clear. Why was this decision made? Why were both guilt and sadness examined in both conditions in the previous study but not this one? Given that it can be difficult to delineate shame and guilt, I would suspect that similar levels of shame and guilt were found in both conditions, which should be reported. 

As we wrote on p. 17, we wanted to facilitate the interpretations of the results. However, we report the full analysis in the Supplement. 

o Why were bootstrap analyses conducted in this study but not the other studies?

We thought that this analysis is necessary because we tested the null effect for shame. However, we decide to remove the bootstrap analyses from the current version of the manuscript. 

o Why were confounds relating to time and distraction not examined in this study? 

We did not examine the time and distraction confounds to keep the study simple. Moreover, because we tested confounds in Study 2, in Study 3, we focused solely on the different effect of pain observation on guilt and shame. 

General Discussion

Does the moral masochism mechanism extend to vicarious pain, or is it specific to self-inflicted pain?

Offer an interpretation of why sadness may have decreased for the experience of vicarious pain, and why this may not have been the case when compared to self-inflicted pain in previous studies. Similarly, offer interpretation of why shame did not decrease. For example, it may be because shame focused on the self, and thus observing pain would not be relevant in reducing this self-focus. 

Discussion of using single items to measure the constructs should be discussed in the limitations. Additionally, it should be noted that, given the conceptual confusion that often occurs between shame and guilt, it is possible that the method of assessing shame and guilt in this study did not server to adequately distinguish between these two constructs. 

We thank you for the reviewer’s suggestions. We followed them and rewrote the general discussion section. 

The implications linking the findings to aggression are unclear. 

We removed the paragraph related to aggression. Instead we added a conclusion paragraph. 

Grammar, formatting, and APA issues:

The paper has numerous grammatical and punctuation errors throughout the manuscript that significantly impede the clarity of this manuscript. A thorough proofreading would be necessary prior to resubmission. 

Change spelling of “behaviour” to “behavior” throughout manuscript.

“CI” should not be italicized. 

As noted in the submission guidelines, footnotes are not permitted.

The headings do not follow proper APA format.

In study 3, the in-text citations are not written in the proper order, as per proper APA formatting.

We proofread the paper with the help of professional proofreading service. We also corrected all the issues referring to the APA format. 

RESPONSES TO ISSUES RAISED BY REVIEWER 2: 

Main points:

1. One of the main methodological concerns is the sample size of each study. The authors provide post hoc power calculations, but it is unclear how they arrived at these numbers or what they mean. Please provide a more detailed account of the power that was achieved by the current sample per study. Please follow the steps outlined by Daniel Lakens here and explain how likely it was to observe a significant effect, given your sample, and given an expected or small effect size and report all parameters entered in G*Power: http://daniellakens.blogspot.com/2014/12/observed-power-and-what-to-do-if-your.html. 

 a. It is a bit unclear why the authors report f effect sizes when their main analyses are repeated measures ANOVAs. 

Based on the reviewer’s suggestion, we provide more information about the effects found in the present studies. Specifically, instead of post-hoc power calculations, we run sensitive power analysis for each study. For example, now we write that for Study 1, our sample size provides 0.80 power for the detection of an effect size of f2 = .21. Also, we report f effect size because we run power analyses for interactions effects. 

2. A conceptual issue relates to the idea that “mere observation of others in pain produces a full experience of pain in the observer” (p. 4). That statement seems hyperbolic and scientifically incorrect based on the research cited. It is true that observing someone else in pain activates similar neural regions than direct experience of pain, but it is also clear that the actual experience of observing someone else in pain and experiencing pain oneself is qualitatively quite different. For instance, no one would confuse whether they experienced pain themselves or they saw someone else experiencing pain. Moreover, the authors did not assess in any way what emotional reaction the participants had to the video of someone else in pain, how “painful” they experienced watching it, or how they were affected by it. In my mind, these are crucial variables to test the hypothesis the authors put forward. Could the authors comment? 

 a. Similarly, the authors keep speaking of “experiencing vicarious pain” without assessing it and is therefore speculative. I would prefer if the authors change this to “observing someone else in pain” or “vicarious pain perception” throughout the whole manuscript and in the title as this more clearly describes the manipulation which was employed.

We much appreciate the reviewer’s suggestions. In the revised version of the manuscript, we change term the “vicarious pain experience”, and instead we use term “pain observation”, “watching other people in pain” or “observe another’s pain”.

3. There are several grammatical and language errors in the text, and the manuscript would profit from professional academic proofreading. 

As mentioned before, the manuscript was proofread by professional service. 

Minor points:

1. P. 2 “pain observation affected participants’ feelings of guilt” Please state the direction of the effect

2. P. 6 “shame and pain experience share the same neural underpinnings” What does that mean and why does that mean it would not be affected?

3. P. 7 “would be observed after watching an exciting and neutral movie clip” Why not use an arousing negative clip as a comparison? It would make the claim stronger that the guilt reduction is in response to pain alone. 

4. P. 8 “asked to write about the last time” How long were the participants required to write? Did all participants write for the required time? 

5. P. 8 Please include a paragraph on the actual design of the study. All studies employed (at least in part) a between-subjects design yet this is not explicitly stated anywhere. Also, the n of each group is not stated or how participants were randomized. Lastly, please refer to between-subject levels as “groups” rather than “conditions” throughout the manuscript.

6. P. 9 “they were told to put their stories in an envelope” What was the cover story and what did the participants think would happen with the stories? Were they told that they were kept confidential? 

7. P. 9 “were measured with two items” Please state the exact wording, scale and labels of the items used.

8. P. 9 Please include a separate paragraph one the exact statistical analyses that were carried out per study, including the dependent and independent variables, their levels and the used alpha-level. 

9. P. 10 “we conducted an ANOVA” Please specify what type of ANOVA (in this case a repeated measures ANOVA)

10. P. 11 “the participants’ level of sadness decreased after the pain manipulation” This is a very surprising finding and definitely requires further elaboration in the discussion. Earlier research shows that watching someone else in pain elicits unpleasant emotions and it is likely that one would feel sad for someone else in pain (especially with high empathy levels). So how do you explain a reduction in sadness when observing someone else in pain?

11. P. 11 In general, there are quite a few post-hoc comparisons. Did the authors employ any post-hoc corrections such as Holm-Bonferroni to correct for multiple testing? This is especially relevant since the studies are likely underpowered. 

12. P. 11 “apart from inducing sadness in this condition, we also elicited guilt” Please elaborate. What do you mean? 

13. P. 11 “(Bastian et al., 2011 (…)” bracket missing

14. P. 13 “We changed the way guilt and sadness was measured” Why did the authors choose to change their measurements throughout the studies? If they want to present it as a replication of Study 1 it seems counterintuitive to change the methodology. Please provide arguments for doing so.

15. P. 13 “was measured with four items” How were these items combined in a single outcome variable? Since this is an entirely new measure, please provide psychometric properties (e.g. Cronbach’s Alpha).

16. P. 14 Did you assess perceived pain intensity or empathy levels at any point? 

17. P. 16 “past studies proved” Please avoid the word “proved” in scientific writing.

18. P. 18 “Finally, using Bayesian interference” The authors introduce a whole new set of statistical analyses here. Please be consistent throughout your studies in the analyses employed and if not, provide a clear rationale for changing method of analysis.

19. P. 22 “might help people deal with aversive feelings of guilt.” The notion that people watch violent shows because of feelings of guilt is highly speculative and far-fetched. I would reduce this last paragraph. Instead, I would be interested in practical or clinical implications of this research as these are currently missing in the discussion and introduction.

We are thankful for these comprehensive and thoughtful suggestions proposed by the reviewer. In the revised version of the manuscript, we followed most of the reviewer’s suggestions by substantially rewriting or clarifying parts that the reviewer was referring to.

---

## [Decision Letter · Decision Letter 1]

2 Nov 2020

PONE-D-20-01585R1

Pain(less) Cleansing:

Watching Other People in Pain Reduces Guilt and Sadness but Not Shame

PLOS ONE

Dear Dr. Bocian,

Thank you for submitting your manuscript to PLOS ONE. Both reviewers have again evaluated the manuscript. Reviewer 1 appreciates the revisions that you made but lists a number of remaining issues. Reviewer 2 repeats a number of points of his earlier review and asks you to clearly state in a cover letter how you have addressed each of his/her comments. I therefore invite you to revise the paper a second time based on the remaining comments of the reviewers. Please make an extra effort in clearly communicating for each individual point how you addressed that point.

We look forward to receiving your revised manuscript.

Kind regards,

Jan De Houwer

Academic Editor

PLOS ONE

Reviewers' comments:

Reviewer's Responses to Questions

**Comments to the Author**

1. If the authors have adequately addressed your comments raised in a previous round of review and you feel that this manuscript is now acceptable for publication, you may indicate that here to bypass the “Comments to the Author” section, enter your conflict of interest statement in the “Confidential to Editor” section, and submit your "Accept" recommendation.

Reviewer #1: (No Response)

Reviewer #2: (No Response)

2. Is the manuscript technically sound, and do the data support the conclusions?

Reviewer #1: Yes

Reviewer #2: Partly

3. Has the statistical analysis been performed appropriately and rigorously? 

Reviewer #1: I Don't Know

Reviewer #2: Yes

4. Have the authors made all data underlying the findings in their manuscript fully available?

Reviewer #1: Yes

Reviewer #2: Yes

5. Is the manuscript presented in an intelligible fashion and written in standard English?

Reviewer #1: No

Reviewer #2: Yes

6. Review Comments to the Author

Reviewer #1: (No Response)

Reviewer #2: Thank you for again receiving the opportunity to review the revised version of the manuscript entitled “Pain(less) Cleansing: Vicarious Pain Experience Reduces Guilt and Sadness but not Shame”. Unfortunately, the authors did not provide a point-by-point reaction to my reviewer comments. This includes my main comment Nr. 2, and all the minor points. In general, please respond to each point individually, indicate where in the text you made changes responding to it, or provide an argument why you felt no change were necessary. Ideally, if the changes are not too substantial (e.g., rewriting a whole part of the text), please quote the text in the response to the reviewer letter with page and line number. This makes it a lot easier to see how the individual points were handled and saves a lot of time in the review process. I again included my original comments below.

Main points:

1. One of the main methodological concerns is the sample size of each study. The authors provide post hoc power calculations, but it is unclear how they arrived at these numbers or what they mean. Please provide a more detailed account of the power that was achieved by the current sample per study. Please follow the steps outlined by Daniel Lakens here and explain how likely it was to observe a significant effect, given your sample, and given an expected or small effect size and report all parameters entered in G*Power: http://daniellakens.blogspot.com/2014/12/observed-power-and-what-to-do-if-your.html.

a. It is a bit unclear why the authors report f effect sizes when their main analyses are repeated measures ANOVAs.

2. A conceptual issue relates to the idea that “mere observation of others in pain produces a full experience of pain in the observer” (p. 4). That statement seems hyperbolic and scientifically incorrect based on the research cited. It is true that observing someone else in pain activates similar neural regions than direct experience of pain, but it is also clear that the actual experience of observing someone else in pain and experiencing pain oneself is qualitatively quite different. For instance, no one would confuse whether they experienced pain themselves or they saw someone else experiencing pain. Moreover, the authors did not assess in any way what emotional reaction the participants had to the video of someone else in pain, how “painful” they experienced watching it, or how they were affected by it. In my mind, these are crucial variables to test the hypothesis the authors put forward. Could the authors comment?

a. Similarly, the authors keep speaking of “experiencing vicarious pain” without assessing it and is therefore speculative. I would prefer if the authors change this to “observing someone else in pain” or “vicarious pain perception” throughout the whole manuscript and in the title as this more clearly describes the manipulation which was employed.

3. There are several grammatical and language errors in the text, and the manuscript would profit from professional academic proofreading.

Minor points:

1. P. 2 “pain observation affected participants’ feelings of guilt” Please state the direction of the effect

2. P. 6 “shame and pain experience share the same neural underpinnings” What does that mean and why does that mean it would not be affected?

3. P. 7 “would be observed after watching an exciting and neutral movie clip” Why not use an arousing negative clip as a comparison? It would make the claim stronger that the guilt reduction is in response to pain alone.

4. P. 8 “asked to write about the last time” How long were the participants required to write? Did all participants write for the required time?

5. P. 8 Please include a paragraph on the actual design of the study. All studies employed (at least in part) a between-subjects design yet this is not explicitly stated anywhere. Also, the n of each group is not stated or how participants were randomized. Lastly, please refer to between-subject levels as “groups” rather than “conditions” throughout the manuscript.

6. P. 9 “they were told to put their stories in an envelope” What was the cover story and what did the participants think would happen with the stories? Were they told that they were kept confidential?

7. P. 9 “were measured with two items” Please state the exact wording, scale and labels of the items used.

8. P. 9 Please include a separate paragraph one the exact statistical analyses that were carried out per study, including the dependent and independent variables, their levels and the used alpha-level.

9. P. 10 “we conducted an ANOVA” Please specify what type of ANOVA (in this case a repeated measures ANOVA)

10. P. 11 “the participants’ level of sadness decreased after the pain manipulation” This is a very surprising finding and definitely requires further elaboration in the discussion. Earlier research shows that watching someone else in pain elicits unpleasant emotions and it is likely that one would feel sad for someone else in pain (especially with high empathy levels). So how do you explain a reduction in sadness when observing someone else in pain?

11. P. 11 In general, there are quite a few post-hoc comparisons. Did the authors employ any post-hoc corrections such as Holm-Bonferroni to correct for multiple testing? This is especially relevant since the studies are likely underpowered.

12. P. 11 “apart from inducing sadness in this condition, we also elicited guilt” Please elaborate. What do you mean?

13. P. 11 “(Bastian et al., 2011 (…)” bracket missing

14. P. 13 “We changed the way guilt and sadness was measured” Why did the authors choose to change their measurements throughout the studies? If they want to present it as a replication of Study 1 it seems counterintuitive to change the methodology. Please provide arguments for doing so.

15. P. 13 “was measured with four items” How were these items combined in a single outcome variable? Since this is an entirely new measure, please provide psychometric properties (e.g. Cronbach’s Alpha).

16. P. 14 Did you assess perceived pain intensity or empathy levels at any point?

17. P. 16 “past studies proved” Please avoid the word “proved” in scientific writing.

18. P. 18 “Finally, using Bayesian interference” The authors introduce a whole new set of statistical analyses here. Please be consistent throughout your studies in the analyses employed and if not, provide a clear rationale for changing method of analysis.

19. P. 22 “might help people deal with aversive feelings of guilt.” The notion that people watch violent shows because of feelings of guilt is highly speculative and far-fetched. I would reduce this last paragraph. Instead, I would be interested in practical or clinical implications of this research as these are currently missing in the discussion and introduction.

7. PLOS authors have the option to publish the peer review history of their article (what does this mean?). If published, this will include your full peer review and any attached files.

Reviewer #1: No

Reviewer #2: No

---

## [Author Response · Author response to Decision Letter 1]

25 Nov 2020

RESPONSES TO ISSUES RAISED BY REVIEWER 1: 

Introduction

I appreciate the authors’ attempts to clarify links between physical and social pain by adding the “Social Pain” section. However, one issue with linking this to guilt is that social pain is actually more closely linked to shame. Shame is thought to act as a warning signal that our need to belong is threatened and is activated in response to social exclusion. This should be noted in this section, and how this might impact the inferences made about links with shame vs. guilt in study 3. 

We much appreciate this suggestion. However, we believe that evidence showing the link between shame and social pain fits better the section regarding Study 3 since, in this section, we want to underline the differences between shame and guilt. Moreover, the “Social Pain” section aimed to show evidence that people may feel pain without experiencing physical pain instead of presenting evidence for the link between social pain and guilt. Since our manipulation of pain observation was not related to social exclusion and therefore, to social pain, we do feel that it makes no sense to write about the link between guilt and social pain in the “Social Pain” section. Therefore, we removed this part from the “Social Pain” section:

Therefore, on the one hand, it is possible that the guilt-reducing effect of pain found in the past research (Bastian et al., 2011; Inbar et al., 2013) may be triggered by contextual factors (e.g., social exlusion). On the other hand, the guilt-reducing effect of pain may by triggered when people observe another’s pain.

Nevertheless, we recognise the importance of the link between social pain and shame in light of Study 3. Thus, on p. 15-16, we write: 

Research concerning the neural activity of brain regions shows another difference between shame and guilt. Experience of shame is associated with increased activation in the ACC (Krach et al., 2011; Michl, 2014), the same brain region, which is active when people experience physical pain, social pain and when they observe pain in others. Furthermore, research has suggested a specific link between social pain and shame, because social exclusion increases shame and this link is mediated by feelings of being devalued (Robertson, Sznycer, Delton, Tooby & Cosmides, 2018). These results correspond with cross-cultural research which has shown that shame is strongly associated with tracking the magnitude of the devaluative threat in others (Sznycer, Tooby, Cosmides, Porat, Shalvi & Halperin, 2016) or evidence that, people who have experienced shame prefer to be together with others over being alone (Hooge, Breugelmans, Wagemans & Zeelenberg, 2018). 

The sentence discussing participants freezing their hand and inhibition (end of pg. 5, start of pg. 6) is still unclear. 

Thank you for this comment. We tried to make the sentence clearer and now, on p. 5 we write:

For example, participants’ hand muscles potentials are reduced when they see another’s hand being hurt, but such inhibition in the sensory-discriminative component of pain was not found when they saw a needle penetrating a tomato or cotton swab moving over the hand. This muscle inhibition suggests activation of a pain resonance system that extracts essential sensory aspects of the other person’s painful experience (Avenanti, Bueti, Galati, & Aglioti, 2005).

Participants needing to select “agree” in Sona to proceed is a form of written consent. This should be clarified accordingly. 

T

hank you. We clarified this and now, on p. 7 we write: 

We obtained the written consent from participants via Sona platform. Participants proceeded to the study only when they agreed to participate in the study after reading the description of the study, time involved, and data policy.

Study 1

How many people were assigned to each condition?

In each condition we had 20 participants. We have added this information o p. 8. 

Specify that the grocery store condition is the control condition.

Corrected on p. 8.

In the first paragraph on page 9, it states that the other person is presumably a doctor. Is this unknown or is this meant to indicate that the participants are likely to presume it is a doctor?

This information is unknown. We write on p. 8: 

In the movie, participants observed the forearm of an unknown person and another person (presumably a doctor, however this information was unknown) trying to insert a needle into the vein of the examined person. 

The last sentence on page 9 states “(any of the participants guessed the true aim of the study).” Should this say “none” instead of “any”?

Corrected on p. 9.

The fact that similar levels of sadness and guilt were observed in both conditions suggests an issue with the validity of your manipulations and should be noted in the discussion.

We appreciate this suggestion. Now, o p. 11, we write:

On the one hand, this finding is likely because, in addition to inducing sadness in this condition, we also elicited guilt. On the other hand, similar levels of sadness and guilt observed in both conditions might suggest some issues with the validity of manipulation used in Study 1.

Study 2

It is still unclear what is meant by “misattributing unrelated excitation”.

We clarified this, and now, on p. 11, we write: 

Specifically, we investigated an idea that unrelated to the pain observation arousal evoked by intensely exciting stimuli instead of a painful one, might later reduce guilt.

Specify whether participants were randomly assigned and how many participants were assigned to each group.

Corrected. Now, on p. 12, we write: 

Based on a random assignment, we asked participants to recall a time when they felt guilty or sad, and then we asked them to either watch the painful movie clip from Study 1, the exciting movie clip or the neutral movie clip (N ~ 26 for each group).

Study 3

On page 15, it is noted that shame is related to competitive behaviour, which is somewhat misleading. 

Thank you for this suggestion. We removed the phrase “competitive behaviour” and now we write on p. 15: 

Moreover, guilt evolved from a need to avoid harming others (Tangney & Dearing, 2002), while in contrast, shame is a self-focused emotion that is related to needs to prove oneself as acceptable or desirable to others (Gilbert, 2003).

There is a discrepancy in the reasoning behind why guilt but not shame would be reduced in the study. In the Social Exclusion section in the introduction, you discuss activation of the ACC in response to social exclusion and that similar patterns are found both when we witness and when we experience social pain. This is used as a possible explanation for findings linking guilt reduction and contextual factors such as social exclusion. However, in the introduction section for study 3, it discusses how ACC activation is associated with shame but not guilt, and the different neural activation is used to justify why shame is hypothesized to remain the same in the study. 

We much appreciate this comment. After carful read, we notice that the part of the paragraph which mentioned contextual factors was confusing. Therefore, we removed it and now we write on p. 5: 

Overall, the evidence we reviewed suggests that people may feel a pain event without directly experiencing psychical pain. In fact, we have ample evidence showing that the vicarious experience of pain can be generated when people observe another’s pain. Therefore, peoples’ physiological response might be similar to, but qualitatively different from, physical pain, or people might report feeling pain themselves when seeing others in pain (Avenanti, Pauello, Bufalari, & Aglioti, 2006; Jackson, Rainville, & Decety, 2006; Lamm, Nusbaum, Meltzoff, & Decety, 2007; Osborn & Derbyshire, 2010).

In the second paragraph on page 15, it is noted that shame is elicited more often by other people, but then a few sentences later it is noted that guilt arises from interpersonal transactions. 

Thank you for this comment. We clarified the paragraph about the differences between guilt and shame and now we write on p. 15: 

Cross-cultural studies have found that participants reported that their shame experiences were elicited more often by other people or external sources than by guilt experiences (Wallbott & Scherer, 1995; see also Fontaine, Luyten, de Boeck, & Corveleyn, 2006). Further, research has demonstrated that when people experience shame, they tend to focus on themselves, but when they experience guilt, they tend to focus on their behaviour (Niedenthal, Tangney, & Gavanski, 1994). Finally, shame arises when adverse events are attributed to one’s stable, global self while guilt arises when adverse events are attributed to unstable, specific aspects of the self (Tracy & Robins, 2004). Moreover, guilt evolved from a need to avoid harming others (Tangney & Dearing, 2002), while in contrast, shame is a self-focused emotion that is related to needs to prove oneself as acceptable or desirable to others (Gilbert, 2003).

In the last paragraph of the introduction to study 3 (pg. 16), the hypothesized outcomes are discussed in terms of their relation with the ACC (e.g., “Therefore, we assume that on the one hand shame induction will recruit the ACC…”). This suggests the study will measure ACC activation, which it did not. Thus, these hypotheses should be reworded. 

Following the reviewer’s suggestion, we reworded this part. No, we write on p. 16:

Overall, presented evidence suggests that experiences of shame, physical pain, and social pain share the same neural basis because they are all associated with increased activation in the ACC (Eisenbeger, 2003; Krach et al., 2011; Michl, 2014; Rainville et al., 1997; Sawamoto et al., 2000). Therefore, it is plausible that shame levels would remain intact after pain observation because the same brain regions process both shame and observation of another person in pain (e.g., ACC). On the other hand, discovered in Study 1 and Study 2 the guilt-reducing effect of pain observation may occur because different brain regions process guilt (e.g., OFC) and these brain regions are not involved in pain processing.

Specify if participants were randomly assigned and how many participants were assigned to each group.

Corrected. Now we write on p. 17: 

Thus, based on a random assignment, we asked participants to recall and describe as precisely as possible a particular event from their life when they either: 1) through their own fault, acted against their own rules, and they caused harm to another person by their fault (the guilt condition, N = 30) or 2) their reputation was damaged, and they could not to change the negative impression that they caused (the shame condition, N = 30).

The statement that the results of study 3 suggest shame my have a weaker association with self-punishment (pg. 18) does not really fit with this study considering the focus was on vicarious pain.

Thank you. We changed self-punishment with the vicarious pain experience. No, we write on p. 18: 

The results of Study 3 corroborated findings from Studies 1 and 2 and provide additional evidence that watching other people in pain reduces feelings of guilt. Additionally, we found that shame remained intact after participants watched another person in pain. This pattern of results suggests that watching other people in pain influences levels of guilt and shame differently when a target emotion is recalled. The results of Study 3 also suggest that shame, as opposed to guilt, might show weaker associations with the vicarious pain experience.

General Discussion

The first sentence of the discussion (pg. 18) states the studies aimed to look at associations between guilt and self-punishment, however, as previously noted, this was not assessed in these studies.

Thank you. We corrected this sentence and now we write on p. 19: 

In this paper, we sought to contribute to the scarce research on the guilt-reducing effect of pain (Bastian et al., 2011; Inbar et al., 2013; van Bunderen & Bastian, 2014) by finding evidence for a specific association between guilt and mere observation of others in pain.

It is unclear how observing pain could have increased activation of the ACC (pg.20). If this were the case, would we not expect shame levels to also increase?

Thank you. We removed the word “increase” and now we write on p. 20: 

Relying on these premises, on the one hand, we assumed that watching other people in pain would recruit the ACC, which past studies have proposed to be an important cortical brain region responsible for pain perception (Casey, Minoshima, Morrow, & Koeppe, 1996; Becerra et al., 1999; Talbot et al., 1991). On the other hand, shame also recruits the ACC (Michl, 2014), therefore, we assumed that shame induction would recruit the ACC and further pain observation sustained this activation. Obviously, these results should be treated with great caution because we did not test activation of the ACC and other brain regions directly involved in guilt and shame processing.

Minor points:

Although substantially improved, there are still several spelling and grammatical errors throughout. The manuscript would benefit from a careful proofreading. 

We believe that some spelling issues are because the text was proofread according to British English rules. However, we again carefully proofread the new version of the manuscript.

Pg. 4 “…and in addition, such emotions as sadness and shame.” should be reworded

We reworded the sentence. Now, on p. 4 we write: 

Specifically, we aimed to examine if observing others’ pain impacts guilt. Furthermore, we tested whether observing others’ pain would affect sadness and shame as well.

Pg. 7, remove the sentence “In this article, we report all measures, all manipulations, and any data exclusions.”

We disagree with the reviewer that this sentence should be removed. This kind of statement is related to guidelines proposed by the open science movement since it improves the transparency of the paper and also states what kind of practices were used during the data analyses.

The verb tense in the description of guilt and sadness (pg. 9) is inconsistent

Thank you. We corrected this issue. Now on p. 9 we write: 

By using a scale from 1 = very slightly or not at all to 5 = extremely, participants indicated to what extend they felt guilty and upset. 

RESPONSES TO ISSUES RAISED BY REVIEWER 2: 

Main points:

1. One of the main methodological concerns is the sample size of each study. The authors provide post hoc power calculations, but it is unclear how they arrived at these numbers or what they mean. Please provide a more detailed account of the power that was achieved by the current sample per study. Please follow the steps outlined by Daniel Lakens here and explain how likely it was to observe a significant effect, given your sample, and given an expected or small effect size and report all parameters entered in G*Power: http://daniellakens.blogspot.com/2014/12/observed-power-and-what-to-do-if-your.html. 

 a. It is a bit unclear why the authors report f effect sizes when their main analyses are repeated measures ANOVAs. 

Based on the reviewer’s suggestion, we provide more information about the effects found in the present studies. Specifically, instead of post-hoc power calculations, we run sensitive power analysis for each study. For example, now we write that for Study 1, our sample size provides 0.80 power for the detection of an effect size of f2 = .21. Also, we report f effect size because we run power analyses for interactions effects. 

2. A conceptual issue relates to the idea that “mere observation of others in pain produces a full experience of pain in the observer” (p. 4). That statement seems hyperbolic and scientifically incorrect based on the research cited. It is true that observing someone else in pain activates similar neural regions than direct experience of pain, but it is also clear that the actual experience of observing someone else in pain and experiencing pain oneself is qualitatively quite different. For instance, no one would confuse whether they experienced pain themselves or they saw someone else experiencing pain. Moreover, the authors did not assess in any way what emotional reaction the participants had to the video of someone else in pain, how “painful” they experienced watching it, or how they were affected by it. In my mind, these are crucial variables to test the hypothesis the authors put forward. Could the authors comment? 

We much appreciate this comment. After a careful read of the manuscript, we decide to rewrite the parts of the manuscript regarding the vicarious pain experience and remove parts which were unclear as the one mentioned by the reviewer. Specifically, we added the section about the social pain o p 4. Moreover, now on p. 5-6, we write:

From a neurological standpoint, pain is multidimensional and covers three broad components: the sensory-discriminative component (the intensity of pain and its bodily location), the affective component (the experience of the unpleasantness) and the cognitive component (de Vignemont & Jacob, 2012; Zaki, Wager, Singer, Keysers, & Gazzola, 2016). These components, due to the observation of another’s pain, might be activated and may produce an experience similar to the experience of physical pain. For example, participants’ hand muscles potentials are reduced when they see another’s hand being hurt, but such inhibition in the sensory-discriminative component of pain was not found when they saw a needle penetrating a tomato or cotton swab moving over the hand. This muscle inhibition suggests activation of a pain resonance system that extracts essential sensory aspects of the other person’s painful experience (Avenanti, Bueti, Galati, & Aglioti, 2005). A different study found activity in the primary somatosensory cortex of participants who observed the needle penetrating the hand (Bufalari, Aprile, Avenanti, Di Russo, & Aglioti, 2007). This evidence suggests that observing which part of the body was hurt and how intensively it was hurt might trigger vicarious pain in the observer via activation of the sensory-discriminative component of pain. 

Vicarious pain experience can be triggered by the affective component of pain, too. Research has demonstrated that the affective component of pain was activated when participants were viewing facial expressions of pain (Botvinick et al., 2005). Similar activation of the affective component of pain was found when participants observed a signal indicating that their loved one receives pain stimulus (Singer et al., 2004). Moreover, we have evidence suggesting that the activation of affective component of pain can be modulated by a wide range of personal and contextual factors, such as the moral behaviour of the person in pain (Singer et al., 2006), his or her medical expertise (Cheng et al., 2007) or pain resulting from medical treatment (Lamm, Batson, & Decety, 2007). 

Cumulatively, research on social and vicarious pain experience suggests that people might report feeling the pain without being physically hurt. Because past evidence has suggested that after the painful experience participants’ levels of guilt decreased after the painful experience (Bastian et al., 2011; Inbar et al., 2013), we propose that observing another’s pain might generate in the observer a superficially similar but qualitatively different kind of painful experience and therefore generate the guilt-reducing effect of pain.

 a. Similarly, the authors keep speaking of “experiencing vicarious pain” without assessing it and is therefore speculative. I would prefer if the authors change this to “observing someone else in pain” or “vicarious pain perception” throughout the whole manuscript and in the title as this more clearly describes the manipulation which was employed.

We much appreciate the reviewer’s suggestions. In the revised version of the manuscript, we change term the “vicarious pain experience”, and instead we use term “pain observation”, “watching other people in pain” or “observe another’s pain” were possible. However, we believe that in some parts of the manuscript we should stick to the term “vicarious pain experience” or “vicarious experience of pain” because these are terms used in the literature describing effects of the pain observation in others.

3. There are several grammatical and language errors in the text, and the manuscript would profit from professional academic proofreading. 

The manuscript was proofread by professional service with the help of two British English native speakers.

Minor points:

1. P. 2 “pain observation affected participants’ feelings of guilt” Please state the direction of the effect

Now we write on p. 2:

Finally, in Study 3 (N = 60), pain observation lowered participants’ feelings of guilt but not their feelings of shame.

2. P. 6 “shame and pain experience share the same neural underpinnings” What does that mean and why does that mean it would not be affected?

This statement is not present anymore in the revised version of the manuscript. However, we explain the difference in the neural activity of brain regions regarding shame, guilt and pain on p. 15 and 16. Also, we added a whole new section about social pain on p. 4

3. P. 7 “would be observed after watching an exciting and neutral movie clip” Why not use an arousing negative clip as a comparison? It would make the claim stronger that the guilt reduction is in response to pain alone. 

We much appreciate this suggestion. At the time of planning experiments, we had access to arousing clip only. Also, we did not think of that the arousing negative clip could be a better option for our claim. Nevertheless, we agree with the reviewer’s suggestion that use of arousing negative clip would make our claim stronger.

4. P. 8 “asked to write about the last time” How long were the participants required to write? Did all participants write for the required time? 

We did not give participants any time restrictions regarding the recall task. Therefore, we did not measure how long participants wrote their memories.

5. P. 8 Please include a paragraph on the actual design of the study. All studies employed (at least in part) a between-subjects design yet this is not explicitly stated anywhere. Also, the n of each group is not stated or how participants were randomized. Lastly, please refer to between-subject levels as “groups” rather than “conditions” throughout the manuscript.

Following the reviewer’s suggestion, we added information about the between-subjects design in the results sections for all three studies. We also changed groups for conditions. 

6. P. 9 “they were told to put their stories in an envelope” What was the cover story and what did the participants think would happen with the stories? Were they told that they were kept confidential? 

The cover story for the experiment was that we are interested in how self-reflection can help people cope with negative life events. There was no cover story for the envelope. We wanted to assure participants that their stories are safe, anonymous and confidential even though we did not say that explicitly. 

7. P. 9 “were measured with two items” Please state the exact wording, scale and labels of the items used.

We added exact wording, scale and labels. On p. 9 we write: 

Guilt and sadness were measured with two items from the 20-item Positive and Negative Affect Schedule (PANAS; Watson, Clark & Tellegen, 1988). By using a scale from 1 = very slightly or not at all to 5 = extremely, participants indicated to what extend they felt guilty and upset.

8. P. 9 Please include a separate paragraph one the exact statistical analyses that were carried out per study, including the dependent and independent variables, their levels and the used alpha-level. 

Information about the detailed statistical analyses that were carried out per study is provided in the results sections.

9. P. 10 “we conducted an ANOVA” Please specify what type of ANOVA (in this case a repeated measures ANOVA)

On p. 19 we write: 

To test the hypothesis that guilt may be reduced by observing another’s pain, we conducted a mixed ANOVA analysis in a 3 (Condition: Guilt vs. Sadness vs. Control) x 2 (Guilt: Time 1 vs. Time 2) design with the first factor as between-subjects and the second factor as within-subjects.

10. P. 11 “the participants’ level of sadness decreased after the pain manipulation” This is a very surprising finding and definitely requires further elaboration in the discussion. Earlier research shows that watching someone else in pain elicits unpleasant emotions and it is likely that one would feel sad for someone else in pain (especially with high empathy levels). So how do you explain a reduction in sadness when observing someone else in pain?

As we stated in the paper, we recognise that the results we found are challenging and puzzling. First on p. 11 we write:

In contrast to past evidence that sadness does not motivate people to seek self-punishment (Inbar et al., 2013), we found that observing another’s pain affected participants’ feelings of sadness. This result is somewhat unexpected and challenging to explain. On the one hand, unlike past research (Inbar et al., 2013), participants in our study could not decide how much pain they wanted to inflict on themselves. Instead, they had to observe another person in pain. Thus, this difference might explain why the participant’s sadness was affected by pain observation manipulation. On the other hand, sadness might fade away with time or because of the intense arousal triggered when participants were watching another person in pain. We sought to examine this alternative explanation in Study 2.

Then, on p. 19-20, we write: 

The original contribution of these studies to the current literature is threefold: First, we contribute to the past research on the guilt-reducing effect of pain by presenting a novel approach to studying how pain observation affects guilt (Bastian et al., 2011; Inbar et al., 2013). Second, in Studies 1 and 2, we found evidence that sadness also decreased after observing another’s pain. Research by Inbar et al. (2013) showed that guilt motivates people to self-punishment more strongly than sadness. In the authors’ study, in all three conditions (guilt, sadness, and control), participants inflicted electroshocks on themselves, but in the guilt condition, participants gave themselves stronger shocks than in the other two conditions. The analysis of guilt at Time 1 and Time 2 showed that stronger shocks were associated with more alleviation of feelings of guilt (Inbar et al., 2013). Unfortunately, there is no information about levels of sadness and the impact that electric shocks had on this emotion. 

It is at least plausible that sadness is also affected by painful experience but in a weaker way than guilt. We found some support for this hypothesis in Study 1. Both guilt and sadness decreased after pain observation, but the size of the effect was more than twice the size for guilt (d = 1.02) than for sadness (d = .45). However, in Study 2, sadness decreased in both the pain and the control groups, thereby possibly implying that sadness wears off over time. More research about sadness in the context of self-punishment and the experience of pain observation is needed.

11. P. 11 In general, there are quite a few post-hoc comparisons. Did the authors employ any post-hoc corrections such as Holm-Bonferroni to correct for multiple testing? This is especially relevant since the studies are likely underpowered. 

For the post-hoc comparisons, we used Bonferroni or Tamhane test.

12. P. 11 “apart from inducing sadness in this condition, we also elicited guilt” Please elaborate. What do you mean? 

We corrected the wording. Now we write on p.11: 

On the one hand, this finding is likely because, in addition to inducing sadness in this condition, we also induced guilt.

13. P. 11 “(Bastian et al., 2011 (…)” bracket missing

Corrected. 

14. P. 13 “We changed the way guilt and sadness was measured” Why did the authors choose to change their measurements throughout the studies? If they want to present it as a replication of Study 1 it seems counterintuitive to change the methodology. Please provide arguments for doing so.

We agree that probably it would be better to use the same measure of guilt and sadness. However, because we did not change the way of how we induced these two emotions, we decided to change the way of how we measured them. Therefore, we wanted to increase the external validity of Study 2.

15. P. 13 “was measured with four items” How were these items combined in a single outcome variable? Since this is an entirely new measure, please provide psychometric properties (e.g. Cronbach’s Alpha).

On p. 12-13 we write: 

Guilt was measured with two items: “guilt” and “remorse”. Participants indicated to what extent they feel each emotion by using a 10-point scale from 1 = I do not feel it at all to 10 = I feel it extremely strongly (Time 1, � = 0.50, M = 3.72, SD = 2.13; Time 2, � = 0.80, M = 3.26, SD = 2.27).

Sadness was measured with four items: “depression”, “sadness”, “misery”, and “breakdown”. Participants rated how they feel each emotion by using a 10-point scale from 1 = I do not feel it at all to 10 = I feel it extremely strongly (Time 1, � = 0.71, M = 2.57, SD = 1.53; Time 2, � = 0.46, M = 2.04, SD = 1.08).

16. P. 14 Did you assess perceived pain intensity or empathy levels at any point? 

Yes, we did. The analyses are provided in the Supplement:

In study 1, we additionally measured how painful the blood collection was according to the participants’ opinion and how much they identified with the protagonist. Participants were asked to report on 5-point scale their judgment from 1 = not at all to 4 = extremely. One-way ANOVA showed no differences between all three conditions for both pain perception, F(2,57) = 1.38, p = .259, and identification with the actor, F(2,57) = 1.03, p = .365.

In study 2, we additionally measured how painful the blood collection was according to the participants’ opinion and how much they identified with the protagonist. Participants were asked to report on 5-point scale their judgment from 1 = not at all to 5 = extremely. One-way ANOVA showed no differences between all conditions for both pain perception, F(2,150) = .32, p = .730, and identification with the actor, F(2,150) = .67, p = .513.

In study 3, we additionally measured how painful the blood collection was according to the participants’ opinion and how much they identified with the protagonist. Participants were asked to report on 5-point scale their judgment from 1 = not at all to 4 = extremely. T-Test analysis showed no differences between the conditions for both pain perception, t(58) = .22, p = .825, and identification with the actor, t(58) = .40, p = .692.

17. P. 16 “past studies proved” Please avoid the word “proved” in scientific writing.

Thank you for this suggestion. We removed word “proved” from the revised version of the manuscript. 

18. P. 18 “Finally, using Bayesian interference” The authors introduce a whole new set of statistical analyses here. Please be consistent throughout your studies in the analyses employed and if not, provide a clear rationale for changing method of analysis.

After considering the reviewer's suggestion, we decide to remove Bayesian interference analysis from the manuscript.

19. P. 22 “might help people deal with aversive feelings of guilt.” The notion that people watch violent shows because of feelings of guilt is highly speculative and far-fetched. I would reduce this last paragraph. Instead, I would be interested in practical or clinical implications of this research as these are currently missing in the discussion and introduction.

We much appreciate this suggestion. We removed from the manuscript the part about violence and instead added a section “Conclusion”. Now on p. 21 we write:

By systematically examining whether watching other people in pain reduces feelings of guilt, this research provides additional support for the theories that argue that guilt leads to a need for suffering (Bastian et al., 2011; Baumeister et al., 1994; Freud, 1924/1953; Inbar et al., 2013). The results of this research suggest that the guilt-reducing effect of pain occurs even without the experience of physical pain. Therefore, we conclude that merely observing other people in pain is sufficient to soothe our feelings of guilt.

---

## [Editor Report · Decision Letter 2]

10 Dec 2020

Pain(less) Cleansing:

Watching Other People in Pain Reduces Guilt and Sadness but Not Shame

PONE-D-20-01585R2

Dear Dr. Bocian,

We’re pleased to inform you that your manuscript has been judged scientifically suitable for publication and will be formally accepted for publication once it meets all outstanding technical requirements.

Kind regards,

Jan De Houwer

Academic Editor

PLOS ONE
---

## [Editor Report · Acceptance letter]

16 Dec 2020

PONE-D-20-01585R2 

Pain(less) Cleansing: Watching Other People in Pain Reduces Guilt and Sadness but Not Shame 

Dear Dr. Bocian:

I'm pleased to inform you that your manuscript has been deemed suitable for publication in PLOS ONE. Congratulations! Your manuscript is now with our production department. 

Kind regards, 

on behalf of

Dr. Jan De Houwer 

Academic Editor

PLOS ONE